# The Possible Outcomes of Poor Adherence to Conventional Treatment in Patients with X-Linked Hypophosphatemic Rickets/Osteomalacia

Hiroaki Zukeran [1,*], Kento Ikegawa [1], Chikahiko Numakura [2] and Yukihiro Hasegawa [1]

[1] Division of Endocrinology and Metabolism, Tokyo Metropolitan Children Medical Center, 2-8-29 Musashidai, Fuchu-shi 183-8561, Tokyo, Japan
[2] Department of Pediatrics, Yamagata University School of Medicine, 2-2-2 Iidanishi, Yamagata-shi 990-9585, Yamagata, Japan
* Correspondence: hzukeran0325.63@gmail.com; Tel.: +81-42-300-5111

**Highlights:**

**What are the main findings?**

- Adherence to conventional treatment is essential but challenging for patients with XLH.
- Conventional treatment should continue even after XLH-children have stopped their growth.

**What is the implication of the main findings?**

- Burosumab is a novel treatment strategy for pediatric patients with poor adherence to conventional treatment.

**Abstract:** X-linked hypophosphatemic rickets/osteomalacia is an inherited disease caused by the loss of function in *PHEX*. Elevated plasma FGF23 in patients with XLH leads to hypophosphatemia. The conventional treatment for XLH, consisting of oral phosphate and active vitamin D, is often poorly adhered to for various reasons, such as the requirement to take multiple daily doses of phosphate. Burosumab, an anti-FGF23 antibody, is a new drug that directly targets the mechanism underlying XLH. We report herein three adult patients with poor adherence to the conventional treatment. In Patient 1, adherence was poor throughout childhood and adolescence. The treatment of Patients 2 and 3 became insufficient after adolescence. All of the patients suffered from gait disturbance caused by pain, fractures, and lower extremity deformities early in life. We prescribed burosumab for the latter two patients, and their symptoms, which were unaffected by resuming conventional treatment, dramatically improved with burosumab. Maintaining adherence to the conventional treatment is crucial but challenging for patients with XLH. Starting burosumab therapy from childhood or adolescence in pediatric patients with poor adherence may help prevent the early onset of complications.

**Keywords:** X-linked hypophosphatemic rickets/osteomalacia; conventional treatment; adherence; burosumab





## 1. Introduction

X-linked hypophosphatemic rickets/osteomalacia (XLH) is an X-linked dominant disorder caused by loss of function in the phosphate-regulating gene with homologies to endopeptidase on the X chromosome (PHEX). Inactivating mutations in PHEX raise plasma concentrations of fibroblast growth factor 23 (FGF23), although the mechanism is not fully understood. Excess FGF23 leads to hypophosphatemia by impairing renal phosphate reabsorption, and decreasing 1,25-dihydroxyvitamin D synthesis [1–3].

The conventional treatment, consisting of oral phosphate and active vitamin D, has some efficacy, but most pediatric patients still have lower limb deformities, diminished

growth, and bone or joint pain. Moreover, because the serum phosphate level returns to a low baseline value within a few hours of phosphate intake, patients must take phosphate several times a day, which contributes to poor adherence [4,5].

The effect of the conventional treatment in adults is also insufficient. Adult patients suffer from bone or joint pain and stiffness, fractures, osteoarthritis, and enthesopathy. Moreover, various complications related to the conventional treatment, such as nephrocalcinosis and hyperparathyroidism, may also occur. Owing to the burden of these symptoms, the quality of life (QOL) of patients with XLH is low [6,7].

Burosumab, a human monoclonal anti-FGF23 antibody approved for use in Japan in December 2019, is the first drug that directly targets the underlying mechanism of XLH. Burosumab is administered via subcutaneous injection every four weeks in adult patients, and every two weeks in pediatric patients [2,3]. Randomized controlled trials with pediatric patients with XLH demonstrated that burosumab therapy was more effective than the conventional treatment in improving rickets, lower limb deformity, growth, and mobility [8,9]. Studies with adult patients demonstrated that burosumab therapy also contributed to healing fractures and improving QOL [10–12].

Herein, we reported three male patients with XLH whose conventional treatment during childhood and adolescence was insufficient. Patient 1 showed a serious outcome that may have contributed to poor adherence to conventional treatment. The other patients demonstrated favorable outcomes after receiving burosumab therapy.

## 2. Case Presentation

### 2.1. Patient 1

The patient presented with hypotonia at the age of 1 year and 7 months, and a bone X-ray revealed rachitic changes in his wrists and knees. Laboratory results confirmed the hypophosphatemia (2.7 mg/dL [normal range: 3.9–6.2 mg/dL]), elevated ALP (642 IU/L, IFCC [normal range: 139–471 IU/L]), and a decreased ratio of the maximum tubular reabsorption to glomerular filtration rate (TmP/GFR) (2.0 mg/dL [normal range: 4.5–6.1 mg/dL]). Based on these findings, he was suspected of having XLH. There was no family history of rickets. A gene analysis at the age of 16 confirmed a frameshift mutation in *PHEX* (c.2473insACTC).

The conventional treatment, consisting of oral phosphate (30 mg/kg) and active vitamin D (0.1 µg/kg), was delayed until the age of 3 years. Even after our diagnosis, his parents refused to acknowledge for some time that he had XLH. After he began walking, genu valgum and waddling gait became apparent. At that time, he stopped attending school because of the embarrassment of having short stature and began acting out violently towards his family. Adherence to conventional treatment was extremely poor throughout childhood and adolescence, hence the lower extremity deformities and short stature failed to improve, and scoliosis developed. He had three genu valgum orthopedic surgery between the ages of 12 and 15. During the third surgery at the age of 15, intramedullary nails were inserted into his left femur. However, there was not much improvement, and the waddling gait persisted. His adult height was 140 cm (−5.3 SD [13]). He continued the oral phosphate and active vitamin D therapy without effect until he decided to discontinue the treatment at the age of 25. Figure 1 shows the severe scoliosis and deformities in his extremities at the age of 33.

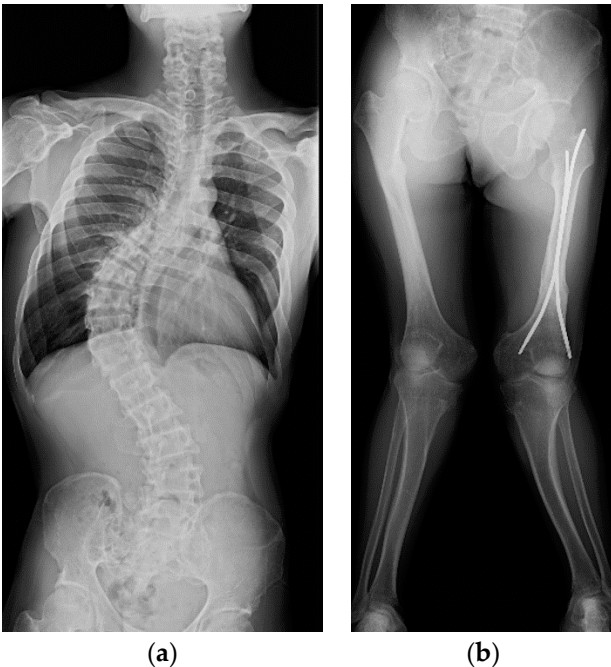

(**a**) (**b**)

**Figure 1.** Bone X-rays at the age of 33. (**a**): Severe scoliosis. (**b**): Malalignment of the lower extremities, including differences in the length of the femurs and bowing of the tibias. (Intramedullary nails were inserted into the left femur).

*2.2. Patient 2*

The patient was referred to our hospital at the age of 11 months because he was unable to crawl. His family history was unremarkable. A physical examination revealed a rachitic rosary and mild thoracic deformity. A laboratory examination demonstrated hypophosphatemia (2.4 mg/dL [normal range: 3.9–6.2 mg/dL]), and a decreased TmP/GFR ratio (2.1 mg/dL [normal range: 4.5–6.1 mg/dL]). No abnormal findings were observed on the other routine tests. Radiographs demonstrated cupping and flaring of the radii and ulnae, indicating XLH. Conventional treatment was started (phosphate 40 mg/kg, active vitamin D 0.1 μg/kg), and his motor development delay and bone deformities improved as a result. At the age of 12, a gene analysis detected a mutation in *PHEX* (IVS9+1 G > A).

His adherence to therapy began worsening from around the age of 17 until he finally stopped visiting the hospital at the age of 21. Although he resumed the phosphate and active vitamin D therapy one year later, he experienced tibial fractures without trauma where technetium-99m bone scan demonstrated abnormal uptake (Figure 2). The lower leg pain was too severe to allow him to walk normally and prevented him from taking a six-minute walk test (6MWT).

Burosumab 0.8 mg per actual weight (1.0 mg per ideal weight) for four weeks was started at the age of 25. The patient reported that, shortly after starting burosumab administration, his pain diminished and was almost completely remitted within six months. One year and two months later, although he was still unable to walk at the average speed for his age, the patient achieved 240 m on a 6MWT and (reference range: 584–686 m [14]). Laboratory data at five months after the start of burosumab therapy found elevated serum inorganic phosphate and TmP/GFR ratio and decreased intact PTH (Table 1). Unfortunately, no blood samples were taken in the first month after burosumab therapy, as recommended in most guidelines [2,3]. A technetium-99m bone scan twelve months after the start of burosumab therapy demonstrated marked improvement (Figure 2).

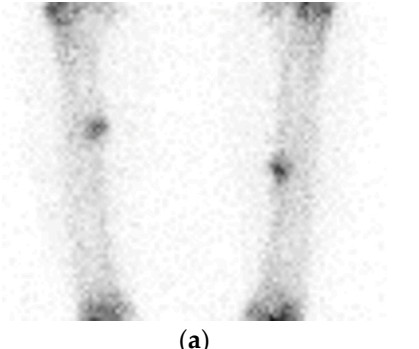
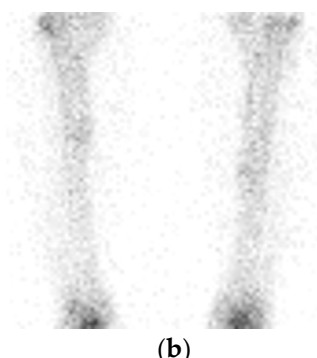

(**a**)  (**b**)

**Figure 2.** Technetium-99m bone scans of the bilateral tibiae. (**a**) Before burosumab therapy: abnormal uptake indicating fractures. (**b**) Twelve months after the start of burosumab therapy: minimal abnormal uptake.

**Table 1.** Laboratory data on Patient 2 before and after the start of burosumab therapy (Tx).

|  |  | At Start of Tx | Five Months after Tx | Reference Range |
|---|---|---|---|---|
| TmP/GFR | (mg/dL) | 0.87 | 1.76 | 2.3–4.3 |
| Serum inorganic phosphorus | (mg/dL) | 1.3 | 2.0 | 2.3–4.5 |
| Serum calcium | (mg/dL) | 9.2 | 9.5 | 8.2–10.4 |
| Serum intact PTH | (pg/mL) | 140.9 | 80.2 | 10.3–65.9 |
| Serum alkaline phosphatase | (IU/L) | 143 | 179 | 110–330 |
| Urinary calcium/creatinine |  | 0.04 | 0.05 | - |

*2.3. Patient 3*

The patient was brought to the hospital at the age of 1 year and 8 months for waddling gait. There was nothing remarkable in his family history. A physical examination revealed genu varum. Laboratory examination found hypophosphatemia (2.2 mg/dL [normal range: 3.9–6.2 mg/dL]), a decreased TmP/GFR ratio (0.6 mg/dL [normal range: 4.5–6.1 mg/dL]), and normal serum 25-hydroxyvitamin D (36 ng/mL). A bone X-ray found wrist and knee rickets. These findings implied XLH, and conventional treatment was begun (phosphate 40 mg/kg, active vitamin D 0.1 μg/kg). The treatment corrected the genu varum, and his gait normalized.

The patient reported difficulty adhering to the regimen of multiple daily phosphate doses during childhood. At the age of 11, with the end of his growth spurt, he discontinued the treatment. Although he continued receiving active vitamin D, the genu varum recurred one year after discontinuing phosphate therapy. At the age of 25, he began experiencing pain in his hip joints. He resumed phosphate therapy at the age of 35 without effect. The pain extended from the hip joints to the lower extremities and lower back.

At the age of 41, he was referred to our hospital. Gene analysis detected a mutation in *PHEX* (c.1735G > A). Six months after the referral, a bone X-ray demonstrated a vertebral compression fracture, and the patient was barely able to walk owing to the severe pain. Burosumab 0.7 mg per actual weight (1.0 mg per ideal weight) for four weeks was prescribed, almost completely eliminating the pain after eight months. One year later, he achieved 364 m on a 6MWT. His serum inorganic phosphorus was higher at one month before the start of burosumab therapy when he was receiving the conventional treatment than five months later. Table 2 shows the sequential changes in the laboratory data, including elevated TmP/GFR. The serum inorganic phosphorus value was highest before burosumab because the sample was obtained after phosphate administration.

**Table 2.** Sequential laboratory data on Patient 3: one month before Tx, at the start of Tx, and five months after Tx.

|  |  | One Month before Tx | At Start of Tx | Five Months after Tx | Reference Range |
|---|---|---|---|---|---|
| TmP/GFR | (mg/dL) | 1.75 | - | 2.11 | 2.3–4.3 |
| Serum inorganic phosphorus | (mg/dL) | 2.9 | 1.7 | 2.7 | 2.3–4.5 |
| Serum calcium | (mg/dL) | 9.7 | 9.6 | 9.6 | 8.2–10.4 |
| Serum intact PTH | (pg/mL) | - | 80.7 | 91.2 | 10.3–65.9 |
| Serum alkaline phosphatase | (IU/L) | 187 | 169 | 152 | 110–330 |
| Urinary calcium/creatinine |  | 0.04 | - | 0.07 | - |

## 3. Discussion

The present report described three adult patients with gait disturbance caused by pain, fractures, and lower extremity deformities early in life. Although the disease expression and outcomes are highly variable, the poor outcomes in present cases may be related to inadequate conventional treatment in childhood and adolescence.

Adherence to the conventional treatment during childhood and adolescence is essential for patients with XLH. Patient 1 suffered from severe lower extremity deformities, and attained a markedly low adult height owing to his poor adherence to therapy throughout childhood and adolescence, and the delayed start of therapy. Before the conventional treatment became available, untreated adult patients were consigned to having severe bone deformities [15–17]. A recent report described a male patient with XLH, which had been misdiagnosed as achondroplasia until the mistake was discovered at the age of 51. Severe deformities developed in the patient's lower extremities and spine, and his final adult height was 127 cm [18].

One of the tremendous burdens of XLH in childhood and adolescence associated with the conventional treatment is the frequent dosing [2,5]. In Patient 2, the worsening of adherence to therapy during adolescence contributed to the formation of fractures and onset of pain in early adulthood. Although no studies have focused on treatment adherence in patients with XLH, the burden of frequent daily phosphate dosing is likely to contribute to discouraging adherence.

Conventional treatment should continue to be administered even after growth is complete. Patient 3 discontinued phosphate after his growth spurt, possibly contributing to his poor prognosis in adulthood. Continuing the conventional treatment into adulthood is controversial because the complications of enthesopathy and osteoarthritis are not alleviated by conventional treatment, and there are concerns about possible side effects, such as hyperparathyroidism and nephrocalcinosis [1,19,20]. Indeed, the 2022 consensus statement [3] does not recommend treating asymptomatic adult patients. However, in our previous report, ten patients with XLH who discontinued the conventional treatment at around the age of 20 became symptomatic (fractures and severe pain) within two to ten years, and resuming the therapy ameliorated their clinical symptoms [21]. These findings suggested that, even if patients are asymptomatic, the symptoms may re-emerge after treatment discontinuation. Continuing the conventional treatment into adulthood can prevent severe complications in patients with child-onset XHL.

Burosumab dramatically improved the pain and gait disorder caused by poor outcomes of the conventional treatment during adolescence and adulthood in Patients 2 and 3, whose symptoms failed to show any improvement after resuming the conventional treatment. Burosumab is recommended for adult patients with persistent bone and/or joint pain, fractures, pseudofractures, or an insufficient response to the conventional treatment [2,3].

On the other hand, the criteria for prescribing burosumab in pediatric patients are unclear. The criteria advocated by one guideline [22] are specific: burosumab is to be administered to pediatric patients with clinical symptoms, such as chronic pain, growth delay, and bone deformity. These criteria do not address poor adherence. Although the 2019 consensus statement [2] recommends burosumab for pediatric patients who are unable

to adhere to conventional treatment, it also states that no conclusive recommendations can be given because the data on the long-term outcomes and cost-effectiveness of burosumab are pending. As mentioned above, conventional treatment is burdensome and difficult for the patient to adhere. To improve adherence in patients with chronic diseases such as XLH, the treatment burden needs continually to be assessed. Depending on the assessment, the treatment may need to be modified [23].

In conclusion, adequate adherence to conventional treatment is crucial but challenging for patients with XLH. Initiating burosumab therapy from childhood in pediatric patients with poor adherence may help prevent complications from appearing early in life. Since XLH is a rare disease and the present report describes only three patients, further long-term studies are required.

**Author Contributions:** Y.H. conceptualized the study. H.Z. wrote the original draft, and Y.H., K.I. and C.N. edited it. All authors have read and agreed to the published version of the manuscript.

**Funding:** This study received no external funding.

**Institutional Review Board Statement:** Ethical review and approval were waived for this study.

**Informed Consent Statement:** Informed consent was obtained from all the subjects mentioned in this report.

**Data Availability Statement:** Not applicable.

**Acknowledgments:** We thank our patients for consenting to participate in this report and James R. Valera for his assistance with editing this manuscript.

**Conflicts of Interest:** The authors declare no conflict of interest.

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
