# Peer review of "The Possible Outcomes of Poor Adherence to Conventional Treatment in Patients with X-Linked Hypophosphatemic Rickets/Osteomalacia"

_endocrines, doi:10.3390/endocrines4010010_

Round 1
Reviewer 1 Report
January 1st 2023
The paper entitles as “Outcomes of poor adherence to conventional treatment in patients with X-linked hypophosphatemic rickets/osteomalacia” by Hiroaki Zukeran et al. reports the adherence of conventional treatment (Vitamin D and phosphate) of three adult patients (males). The paper concludes that maintaining adherence is crucial but challenging due to the adverse effects. However new FGF23 related treatments as burosumab helps preventing the early onset complications, especially relevant in those with poor adherence.
The paper is well written and easy to follow. The research conducted contributes to increase the knowledge of this rare disorder. It is already described that XLH treatment is usually initiated at the time of diagnosis and continued for live long or at least until growth completion. Disease severity is not the same in all patients, and some respond better to treatment than others. The challenges often faced with conventional treatment include variable compliance and poor tolerability including gastrointestinal symptoms. Therefore, is undouble that clinical as well as basic studies are crucial for XLH proper treatment. The major weakness of the paper is the low number of patients, that in part can be explained by the fact that XLH is a rare disorder, however, get conclusions based only in thee individuals is ambitious.
Results are clear and very well presented. Treatment Recommendations for Borosumab based on the product information are: fasting serum phosphate levels should be measured before initiating burosumab, following a 7-day washout period if the patient was previously on conventional treatment (Which is the case). Serum phosphate should be monitored on a 2-weekly basis for the first month, 4-weekly for the following 2 months. This matches in part with the experiment design. An explanation of why phosphorus was not measure one month after the start of Borosumab treatment in patient 2 is needed.
Proper and early diagnose as well as adherence to the conventional treatment during childhood and adolescence are two essential factors for patients with XLH, however Vitamin D and Pi needs daily dose which, as the authors described, is a tremendous burden. In this scenario I think more debate to improve this issue in the discussion section will improve the quality of the paper (Treatment Burden and Treatment Fatigue as Barriers to Health. Bryan W. Heckman et al 2015).
I also recommend the author to add Learning Points section
Minor issues
Reference needs homogenization. 15, 16 and 17 do not have DOI. Reference 2 and 3 are incomplete
Author Response
Thank you very much your constructive comments. We revised our manuscript accordingly. Our replies to the reviewer are in red.
The paper is well written and easy to follow. The research conducted contributes to increase the knowledge of this rare disorder. It is already described that XLH treatment is usually initiated at the time of diagnosis and continued for live long or at least until growth completion. Disease severity is not the same in all patients, and some respond better to treatment than others. The challenges often faced with conventional treatment include variable compliance and poor tolerability including gastrointestinal symptoms. Therefore, is undouble that clinical as well as basic studies are crucial for XLH proper treatment. The major weakness of the paper is the low number of patients, that in part can be explained by the fact that XLH is a rare disorder, however, get conclusions based only in thee individuals is ambitious.
This is an important point we should address. We added “Since XLH is a rare disease and the present report describes only three patients, further long-term studies are required.” in the Conclusion. (lines 208 – 209)
Results are clear and very well presented. Treatment Recommendations for Borosumab based on the product information are: fasting serum phosphate levels should be measured before initiating burosumab, following a 7-day washout period if the patient was previously on conventional treatment (Which is the case). Serum phosphate should be monitored on a 2-weekly basis for the first month, 4-weekly for the following 2 months. This matches in part with the experiment design. An explanation of why phosphorus was not measure one month after the start of Borosumab treatment in patient 2 is needed.
Unfortunately, we did not have a chance to take blood samples from Patient 2 in the first month after burosumab therapy, as recommended in most guidelines. (lines 117 – 118)
Proper and early diagnose as well as adherence to the conventional treatment during childhood and adolescence are two essential factors for patients with XLH, however Vitamin D and Pi needs daily dose which, as the authors described, is a tremendous burden. In this scenario I think more debate to improve this issue in the discussion section will improve the quality of the paper (Treatment Burden and Treatment Fatigue as Barriers to Health. Bryan W. Heckman et al 2015).
Thank you for your important suggestion. As suggested, we mentioned the necessity of assessing the burden on patients and the possible need to modify the treatment to improve adherence in the Discussion section. (lines 201 – 204)
I also recommend the author to add Learning Points section
Thank you for your recommendation. We included three Learning Points. (lines 28 – 34).
Minor issues
Reference needs homogenization. 15, 16 and 17 do not have DOI. Reference 2 and 3 are incomplete
No DOI was available for references 15 -17 because they were old. We have completed the entry for references 2 and 3.
We sincerely hope you will find the revised version satisfactory.
Best regards,
Hiroaki Zukeran, M.D.

Author Response
Thank you very much for your constructive comments. We revised our manuscript accordingly. Our replies to the reviewer are in red.
General concept comments:
A brief search of pubmed reveals no similar case studies, case series, observational reports, or clinical trials of patient outcomes to burosumab therapy after non-adherence to conventional therapy, making this a unique addition to the literature. Additionally, the topic of XLH is of interest to readers of the Journal. My biggest question is what value patient 1 adds to this case series since they did not receive burosumab therapy. The article may be better focused on only the latter 2 patients, with an updated title to reflect outcomes of burosumab therapy after poor adherence to conventional therapy.
- Introduction
- Page 2, line 52-54: This sentence makes it appear as if all 3 patients received burousmb therapy as the authors states that the three male patients “later demonstrated favorable outcomes after….burosumab”. This needs revised for accuracy, or patient 1 can be removed from the manuscript in its entirety.
Thank you for your comments. We would like to show in Patient 1 the serious outcomes which poor adherence to conventional treatment may cause. We revised the last sentence in the Introduction as follows: Patient 1 experienced a serious outcome that may have contributed to poor adherence to conventional treatment. The other patients demonstrated favorable outcomes after receiving burosumab therapy. (lines 61 – 63)
- Case presentation
- Patient 1
- Page 2, Line 64-66 – Why did the parents’ request to delay phosphate and vitamin D?
His treatment was delayed because his parents refused for some time to accept that he had XLH. We explained this in the revised version. (lines 74 – 76)
- Page 2, Line 68 – “His drug adherence…” sounds odd. Consider rewording to “adherence to conventional therapy was extremely poor…”
Thank you for your suggestion. We reworded this sentence as suggested. (line 79)
- Page 3, Figure (b), Line 78-89 – Specify both what age the nails were inserted and during which surgery (1, 2 or 3)
We have added the information requested. (lines 82 – 83)
- Patient 2
- Page 4, Figure 2 and Table 1 – These are both very helpful at quickly comparing the results of burosumab therapy. One suggestion for Table 1 is to include the normal reference range for readers. Another is to specify the timing of the lab draw before thearpy (i.e. 1 month? 5 months? At the start of therapy?)
In response to your comments, we added the reference range to Table 1 and changed “before Tx” to “At the start of Tx”.
- Patient 3
- Page 5, Table 2 – Similar to above, consider addition of normal reference range.
We added the reference range to Table 2.
We sincerely hope you will find the revised version to be satisfactory.
Best regards,
Hiroaki Zukeran, M.D.

Reviewer 3 Report
Zukeran et al. reported a case series of three patients diagnosed with XLH who did not comply with conventional therapy and had poor clinical outcomes; the manuscript is well written, but I have some comments for the author:
1. The title stated that the result of poor adherence to conventional therapy, Is this true? Because we know that disease expression is highly variable and that some patients, even with excellent compliance, will have a serious outcome, I would suggest changing the title to the possible outcomes of poor adherence to conventional treatment in patients with X-linked hypophosphatemic rickets/osteomalacia.
2. Include the reference ranges for Po4, ALP, and TmP/GF, as well as the dose of conventional therapy, in the three cases. This will add to the report and help the reader understand that the patients were treated at an adequate dose.
3. Add the reference range to Table 2; it appears that the phosphate level before burosumab is lower than after burosumab; please explain this as it contradicts the text; and what was the ALP level before and after burosumab
4. This is a very strong statement in the discussion, lines 139-140; I believe the disease expression is variable, which could be due to insufficient dose or the disease itself.
5. Please mention in the discussion that the disease expression is variable and the outcome may be variable.
8. In conclusion, please add after the last statement that further long-term studies are required.
Author Response
Thank you very much for your constructive comments. We revised our manuscript accordingly. Our replies to the reviewer are in red.
- The title stated that the result of poor adherence to conventional therapy, Is this true? Because we know that disease expression is highly variable and that some patients, even with excellent compliance, will have a serious outcome, I would suggest changing the title to the possible outcomes of poor adherence to conventional treatment in patients with X-linked hypophosphatemic rickets/osteomalacia.
We appreciate your helpful suggestion. As you point out, their outcomes cannot be attributed solely to poor adherence. We edited the title as you suggested.
- Include the reference ranges for Po4, ALP, and TmP/GF, as well as the dose of conventional therapy, in the three cases. This will add to the report and help the reader understand that the patients were treated at an adequate dose.
Following your suggestion, we added the reference ranges for the biochemical parameters and the dosage for conventional treatment in each patient. (Patient 1: lines 68 – 71, 74 – 75 ) (Patient 2: lines 97 - 98, 100 – 101) (Patient 3: lines 129 – 131, lines 133 – 134) For Patient 2, we deleted the ALP level from the text because it was not outside the normal range.
- Add the reference range to Table 2; it appears that the phosphate level before burosumab is lower than after burosumab; please explain this as it contradicts the text; and what was the ALP level before and after burosumab
We added the reference range and ALP levels to Tables 1 and 2.
In patient 3, the serum inorganic phosphorus value was highest before burosumab therapy because the sample was obtained after phosphate administration. We explained this in the revised version. (lines 149 – 150)
- This is a very strong statement in the discussion, lines 139-140; I believe the disease expression is variable, which could be due to insufficient dose or the disease itself.
- Please mention in the discussion that the disease expression is variable and the outcome may be variable.
Thank you for your remarks. We edited lines 139-140 in the first version. We commented on the disease expression as follows: Although the disease expression and outcomes are highly variable, the poor outcomes in present cases may be related to inadequate conventional treatment in childhood and adolescence. (lines 156 – 158)
- In conclusion, please add after the last statement that further long-term studies are required.
We added as follows: Since XLH is a rare disease and the present report describes only three patients, further long-term studies are required. (lines 208 - 209)
I sincerely appreciate your constructive review and hope you will find the revised version satisfactory.
Best regards,
Hiroaki Zukeran

Round 2
Reviewer 1 Report
Author have improved the quality of the paper. They have added the learning points and justified the controversial points. However, I recommend a deep English revision prior publication. I have some recomendations for the authors bellow
29>Adherence to the conventional treatment is essential but challenging for patients with 29 XLH. Withouth THE or Alternative: conventional treatment adherence is essential but...
31-32 > Conventional treatment should continue to be administered even after patients with XLH diagnosed during childhood have completed their growth. REFRASE. Alternative suggestion: Conventional treatment should continue even after XLH-children have stopped their growth.
68>A laboratory examination demonstrated hypophosphatemia. Sounds weird. Alternative: Laboratory results confirmed the hypophosphatemia
72>Based on these findings, hypophosphatemic rickets was diagnosed. Hipophosphatemic ricket or XLH? Because you can have hipophosphatemic ricket manifestations in other diseases... also I recomend to move that sentence to the end of the paragraph as a conclusion
76>was delayed until age 3 years. Poor english. Rephrase: Was delayed until the ege of three (3 years and 10 months).
75-77 The conventional treatment, consisting of oral phosphate (30 mg/kg) and active vitamin D (0.1 μg/kg), was delayed until age 3 years 10 months at his parents’ request because his parents refused to acknowledge for some time that he had XLH. This sentence is very long and doesn´t sound natural.
83. He had three orthopedic operations for genu valgum between ages 12 and 15 years. This error is constant. In english you say years of age, years old or the age of... Therefore: He had three genu valgum orthopedic SURGERY between the ages of 12 and 15. Same in the next sentence. Change OPERATION for SURGERY
134. Bone X-ray found rickets in his wrists and knees. Bone X-ray found Wrist and knee rickets. Based on these findings, hypophosphatemic rickets was diagnosed. Once again what do you mean? XLH was diagnosed?
144. At age 41 years. Remember: At the age of 41, or at 41 years old
206-207> To improve adherence in patients with chronic diseases like XLH, the treatment burden and capacity of individual patient to adhere to the treatment need continually to be assessed. This sentence sounds very repetitive and not very natural.
Author Response
We sincerely appreciate your helpful suggestions and have revised our manuscript accordingly. Our replies to the reviewer are given in red.
29>Adherence to the conventional treatment is essential but challenging for patients with 29 XLH. Withouth THE or Alternative: conventional treatment adherence is essential but...
We deleted THE from this sentence (line 29).
31-32 > Conventional treatment should continue to be administered even after patients with XLH diagnosed during childhood have completed their growth. REFRASE. Alternative suggestion: Conventional treatment should continue even after XLH-children have stopped their growth.
We revised the sentence as suggested (lines 30-31).
68>A laboratory examination demonstrated hypophosphatemia. Sounds weird. Alternative: Laboratory results confirmed the hypophosphatemia
We reworded this sentence according to your recommendation (lines 66-67).
72>Based on these findings, hypophosphatemic rickets was diagnosed. Hipophosphatemic ricket or XLH? Because you can have hipophosphatemic ricket manifestations in other diseases... also I recomend to move that sentence to the end of the paragraph as a conclusion
Thank you for your remarks. The first time Patient 1 was brought to our hospital, we did not perform a genetic analysis and no family history of XLH was found. However, we suspected that he had XLH based on its frequency. Therefore, we did not remove the sentence but revised it as follows: Based on these findings, he was suspected of having XLH (line 70).
76>was delayed until age 3 years. Poor english. Rephrase: Was delayed until the ege of three (3 years and 10 months).
- At age 41 years.Remember: At the age of 41, or at 41 years old
We reworded these phrases throughout the manuscript as you recommend.
75-77 The conventional treatment, consisting of oral phosphate (30 mg/kg) and active vitamin D (0.1 μg/kg), was delayed until age 3 years 10 months at his parents’ request because his parents refused to acknowledge for some time that he had XLH. This sentence is very long and doesn´t sound natural.
We divided this sentence as follows: The conventional treatment, consisting of oral phosphate (30 mg/kg) and active vitamin D (0.1 μg/kg), was delayed until the age of 3 years. Even after our diagnosis, his parents refused to acknowledge for some time that he had XLH. (lines 73-75)
- 83. He had three orthopedic operations for genu valgum between ages 12 and 15 years. This error is constant. In english you say years of age, years old or the age of... Therefore: He had three genu valgum orthopedic SURGERY between the ages of 12 and 15. Same in the next sentence. Change OPERATION for SURGERY
We corrected this error and changed OPERATION to SURGERY (lines 80-81).
- Bone X-ray found rickets in his wrists and knees. Bone X-ray found Wrist and knee rickets. Based on these findings, hypophosphatemic rickets was diagnosed. Once again what do you mean? XLH was diagnosed?
As we explained in regard to Patient 1, we revised the sentence as follows: These findings implied XLH (line 131).
206-207> To improve adherence in patients with chronic diseases like XLH, the treatment burden and capacity of individual patient to adhere to the treatment need continually to be assessed. This sentence sounds very repetitive and not very natural.
We revised this sentence as follows: To improve adherence in patients with chronic diseases like XLH, the treatment burden needs continually to be assessed (lines 200-202).
We sincerely hope you will find the revised version to be satisfactory.
Best regards,
Hiroaki Zukeran, M.D